# A Novel Summer Camp Integrating Physical, Psychological, and Educational Health in Youth: The THINK Program

**DOI:** 10.3390/nu16121838

**Published:** 2024-06-12

**Authors:** Joseph Bonner, Wei Xiong, Carolina Velasquez, Natasha Nienhuis, Brady Wallace, Alexis Friedman, Debbiesiu Lee, Arlette Perry

**Affiliations:** Department of Kinesiology and Sport Sciences, University of Miami, 5202 SW University Drive, Coral Gables, FL 33146, USA; wxx143@miami.edu (W.X.); cxv308@med.miami.edu (C.V.); natasha.nienhuis21@gmail.com (N.N.); bhw24@miami.edu (B.W.); a.friedman8@umiami.edu (A.F.); debbiesiu@miami.edu (D.L.); aperry@miami.edu (A.P.)

**Keywords:** physical fitness, nutrition knowledge, adolescents, health promotion

## Abstract

Numerous youth summer programs focus upon physical fitness, nutritional health, psychological well-being, or education. Few, however, have integrated all of these elements into a single program. The Translational Health in Nutrition and Kinesiology (THINK) program provides an integrative nutrition and exercise science curriculum that is interfaced with social emotional learning (SEL) and STEM education to enhance healthy behaviors in youth. The purpose of this study was to determine whether the THINK program could improve physical fitness, nutrition habits, SEL, and STEM education in a 6-week summer program covering a 3-year period. Participants from South Miami were enrolled in THINK (*n =* 108, 58 males, 50 females, 12.03 + 0.56 years). Physical fitness assessments, the Positive Youth Development Inventory (PYDI), the Students’ Attitude Towards STEM Survey, and the Adolescent Food Habits Checklist (ADFH) were recorded at baseline and post-testing. Means and standard error values were evaluated for all dependent variables. Paired samples *t*-tests (SPSS version 27) were used to determine changes. Improvements in cardiorespiratory fitness (*p <* 0.001), power (*p <* 0.006), flexibility *(p* < 0.001), agility (*p <* 0.001), muscular endurance (*p <* 0.001), lean body mass (*p <* 0.001), ADFH (*p <* 0.001), and PYDI (*p =* 0.038) were found. An integrative summer fitness program can result in improvements in physical fitness, nutrition habits, and SEL in as little as six weeks.

## 1. Introduction

During the last six decades, fitness levels in youth have declined both in the United States and globally [1]. The prevalence of cardiovascular disease (CVD) risk factors is now found to occur in adults as young as 20 years old. A cross-sectional study of 12,924 young adults (ages 20–44) from 2009 to 2020 found significant increases in type 2 diabetes and obesity rates, with Latino/x and non-Hispanic Black young adults possessing the highest prevalence of CVD risk [2]. As of 2022, rates of obesity among adolescents in the United States have increased to an all-time high of 20.6% [3]. Furthermore, these rates were highest among Latino/x and non-Hispanic Black youth [4]. In Western European countries of similar socioeconomic status, their obesity rates are lower: the United Kingdom (11.3%), Germany (8.49%), the Netherlands (4.75%), and France (4.3%) [3]. The “State of Childhood Obesity Initiative” reported that Florida ranked 14th out of the 50 United States, with 14.6% of children 10–17 years having obesity [5]. Research has shown that obese adolescents experience more adverse medical conditions, including pre-diabetes, hypertension, asthma, and fatty liver disease [6]. Since low levels of physical activity and fitness are associated with numerous adverse health outcomes that worsen with age [7], low physical fitness concomitant with overweight/obesity can have a critically negative impact on overall health and well-being. Furthermore, both overweight/obesity [8] and physical fitness levels track into adulthood [9], which means that overweight/obese youth maintain their relative position as they transition into adulthood. Thus, there is an urgent need to promote healthy behaviors early on among adolescents to reduce CVD risk factors in adulthood. This is especially important for adolescent minority youth who possess the highest rates of adiposity [10] and lowest rates of physical fitness [11].

In the wake of recovery from the COVID-19 pandemic, wherein school closures, reduced physical education classes, greater calorie consumption, and psychological stresses were exacerbated [12,13,14,15,16], many programs developed to offset the detrimental effects of the pandemic on our nation’s physical and mental health and fitness [17,18,19]. While many of these community-based summer programs focused upon physical fitness [20], nutrition habits [21], psychological health [22], and STEM education [23] independently, few had integrated all of the above key elements into a single curriculum.

The Translational Health in Nutrition and Kinesiology (THINK) summer program was born out of the need to develop a more holistic approach to physical, psychological, and educational well-being. This integrative curriculum seeks to empower youth to improve their own personal lifestyle behaviors and inspire them to “THINK” about their personal nutrition and exercise choices. The purpose of this study was to determine whether a pioneering, 6-week summer program could improve physical fitness, social emotional learning (SEL), nutrition behaviors, and STEM education in a single program across a 3-year period.

## 2. Materials and Methods

The attendees were recruited from Miami Dade County Public Schools in South Florida using informational flyers sent to school principals, counselors, and other public administrators. Participants were enrolled on a first come, first serve basis, provided they would be present for the entire 6-week program. This study was conducted in accordance with the Declaration of Helsinki, with approval for this study provided by the University of Miami’s Institutional Review Board (protocol # 20190472). All parents gave written informed consent, with their children providing informed assent to participate in this study. The families of the participants were charged a small registration fee of USD 50 for THINK program t-shirts but were not required to pay for any other program-related fees. During 2019, 2022, and 2023, a total of 147 adolescents were enrolled in the 6-week THINK summer program held on a university campus. Approximately 25% of the adolescents consistently returned to the camp for more than 1 year, and 75% were new participants with each summer cohort. Due to external conflicts (travel plans), personal issues (injury, sickness, and loss of designated driver), and other family circumstances, 121 participants completed baseline and post-testing, which translated to an 8.66% student drop-out rate per year.

Prior to the onset of the program, faculty experts in the Kinesiology and Sport Sciences, Educational and Psychological Studies, and Teaching and Learning departments worked together with graduate assistants piloting several projects that had been published elsewhere [24,25,26]. The lead investigators and their assistants also held an orientation/ training workshop to acquaint counselors with the vision and mission of the program; review procedures for conducting assessments; and inform participants on methods to enhance integration of physical, psychological, and educational well-being. The research team also hosted a parent orientation to review the program’s objectives, content, activities, and curriculum along with the necessary forms needed before participation (consent and assent, code of conduct, medical liability, and medical history). Participants completed both baseline and post-testing data collection, which included demographic information, physical measurements, physical fitness measures, and self-reported surveys/questionnaires.

### 2.1. Program

The THINK program encompassed engaging lessons in nutrition and exercise science interfaced with hands-on laboratory activities to promote STEM learning. Physical activities promoting sportsmanship, camaraderie, and fun were also integrated into the program. Principles of SEL were infused throughout the curriculum to support positive lifestyle behaviors, overcome barriers to exercise, attain personal goals, and inspire participants to improve their physical and mental health. Discussions always began with probing questions that prompted students to think about their answers, promote student engagement, and encourage student interaction. Topics on physical fitness, warm-up/stretching, athletic injuries, the heart, and metabolic fitness were part of the curriculum.

Nutrition education was an important component of the curriculum. Classroom presentations focused upon relevant nutrition topics such as processed versus unprocessed foods, nutrient-dense foods, deciphering macronutrients from micronutrients, alternative plant-based options, hydration needs, and reading labels. During lectures on the heart and metabolic health, the class was introduced to the Mediterranean meal plan and the importance of a well-rounded diet featuring a variety of vegetables and fruits. Although adolescents brought their own lunches daily, investigators facilitated conversations during nutrition lectures about what choices they could make to improve their lunches and other meals to make them healthier. The curriculum also incorporated games and activities that emphasized healthy eating habits. For example, following the nutrition unit, students had to sprint to a shopping bag containing groceries, pick an item from the shopping bag, then sprint to a My Plate poster, and place the selected item under the appropriate food group. Points were awarded for the fastest teams with the most accurate placement of food items. 

Hands-on clinical laboratories were included to enhance STEM learning while enabling students to learn more about their bodies and how it functions from a science-based perspective. These experiences provided opportunities to use a fully equipped university exercise physiology laboratory. Therefore, following informational sessions, the attendees learned how to use goniometers to measure range of motion; calipers to measure skinfold thickness; dynamometers to measure muscular strength; and field tests to measure, agility, aerobic fitness, and/or balance. Laboratory experiences served as a creative forum for reinforcing concepts learned in the classroom setting. Some of the laboratories also featured a competitive element such as “Simon says” wherein students pointed to the correct bone (ulna, femur, or acromium process) or muscle (deltoid, gastrocnemius, or triceps) to stay in the game. Both laboratory experiences and physical activities reinforced health-related themes. 

The SEL component was drawn from the Social Cognitive Theory [27] and grounded in the general conceptual model showing that positive relationships between physical activity and physical fitness translate into a better quality of life and health [27,28]. Methods for promoting and sustaining positive behavioral changes included identifying emotions, making responsible choices, setting up motivational goals, appreciating others, reaching out for help, and empowering youth to make positive lifestyle behavioral changes. The curriculum provided a nurturing, socially positive environment that encouraged youth to “think deeply” about their positive growth and actions in all aspects of their life. This included making healthy decisions in nutrition and exercise, developing confidence and competence in academics, particularly STEM, and promoting communication and self-reflection while building positive relationships. 

### 2.2. Physical Measurements 

Height was recorded via a wall-mounted stadiometer, and weight was measured on an electronic scale. Body fat, muscle mass, and weight were measured via a multi-frequency segmental bioelectrical impedance analysis utilizing an Inbody 570 machine (Seoul, South Korea). The Inbody 570 has been validated as a fairly reliable correlate of underwater weighing [29]. Within adolescent populations, a comparison on Inbody 570 bioelectrical impedance with underwater weighing found fair correlations with UW (*r* = 0.79 for girls and *r* = 0.69 for boys, *p* < 0.10 for both) [29]. Body mass index (BMI) was calculated using the formula weight (kg)/height (m)^2^. Percentiles above or below the 85th percentile were based on the Center for Disease Control’s guidelines [30].

### 2.3. Physical Fitness Measures

Muscular Strength: Grip strength was measured via a digital handheld dynamometer (Camry Scale, El Monte, CA, USA) While in a seated position with their hand bent at a 90-degree angle, participants squeezed the dynamometer as hard as possible using their dominant hand [31]. The use of digital handgrip dynamometers has been validated in adolescent populations. A study involving 338 participants, 7–13 years, found that digital handheld dynamometers possessed a high intraclass correlation for test–re-test reliability in 10–13 years olds (*n* = 169, (*ICC* = 0.98, (0.97–0.98) [32]. The Camry Scale digital handheld dynamometer has also been shown to correlate well with the widely used Jamar Handheld grip dynamometer [33]. 

Cardiorespiratory Fitness: This was evaluated via the 20 m Fitnessgram Pacer Test wherein students ran as many laps as possible within a specified time allotted for each lap until the student could no longer keep pace completing laps [34]. The Fitnessgram Pacer Test has been shown to be a validated measure of estimated aerobic fitness and predicted VO_2_ max [35]. A study comparing 244 boys and girls between 10 and 16 years old demonstrated a significant correlation between predictive VO_2_ max models from the Pacer Test and a graded exercise test to maximum (*r* = 0.75, and the SE of estimate (SEE) was 6.17 mL/kg/min) [35]. 

Muscular Endurance: Abdominal muscular endurance was determined via the “curl-up” test, recommended by the President’s Council of on Physical Fitness and Sports [36]. Regarding reliability, instructor-reported scores for the “Curl-Up Test” have been previously reported at moderate levels (*r* = 0.75 for girls and *r* = 0.80 for boys) [37]. During testing, participants performed as many curl-ups as possible within one minute, with their feet and gluteus region separated by 12 in while in a supine position. 

Power: Lower body power was tested using the Vertec TM Jump Training System (JumpUSA, Sunnyvale, CA, USA). Each adolescent’s standing height with their arm stretched overhead was subtracted from their maximal jump. The jump entailed one step forward and a squat countermovement to preferred depth before jumping [38]. In a concurrent study of 529 adolescents between 10 and 18 years, validation and cross-validation measures of vertical jump height did not differ significantly (*p* > 0.05) [39]. Power estimated from the vertical jump test significantly correlated with vertical jump power recorded from a mechanography system that measured ground reaction forces with two force plates (*r* = 0.91 to 0.93) [39]. 

Flexibility: Range of motion of the lower back and hamstring muscles were measured using a Sit-and-Reach Box (Acuflex I, Novel Products Inc., Rocktown, IL, USA) [40]. Subjects were instructed to sit on the floor with their outstretched legs abutting the Sit-and-Reach Box and asked to slowly move a lever as far forward as possible without bending their knees. The previously reported criterion’s validity for hamstring and lower back flexibility among adolescents using the Sit-and-Reach test has been moderate, with values ranging from *r* = 0.37 to 0.76 [40,41,42]. The test–retest reliability for the Sit-and-Reach test is reported to be high using a goniometer (*r* = 0.88–0.99) [43]. 

Agility: This was evaluated using the 30-foot Eraser Shuttle-Run, using the protocol from the President’s Council on Physical Fitness and Sports [36]. Participants were required to run as quickly as possible to retrieve two small board erasers from the opposite end, one at a time, covering a 30-foot distance. The time was recorded after the second eraser was successfully placed at the starting point. Using test–retest scores, one study of 123 male and female adolescents, 13.6 + 0.8 years, found a non-significant mean difference of 0.1 ± 0.7 s for boys and 0.1 ± 0.8 s for girls (*p* > 0.05) [44].

### 2.4. Self-Reported Questionnaires/Surveys (See Appendix A)

Nutrition Habits: To assess general food habits, the Adolescent Food Habits Checklist (AFHC) [45] was used. The survey addressed areas wherein adolescents are more likely to have some degree of control with respect to the consumption of calorically dense foods, low fat alternatives, fruits and vegetables, and snacking behaviors. This survey involved answering 23 questions on specific dietary practices with an answer of “true”, “false”, or “doesn’t apply to me”. For each question answered true or false, one point was given for a healthy response. For example, answering “true” to question four: “I make sure I eat at least one serving of fruit a day” would yield one point for a healthy response. The final score after tallying all healthy responses was incorporated into the formula below to determine the overall AFHC score:

AFHC score = no of ‘healthy’ responses × (23/no of items completed) [45].

Scores have ranged from a minimum of 0 to a maximum score of 23. The AFHC has been considered a reliable and valid self-reported survey for assessing dietary habits in adolescents. An investigation that included 1882 adolescents between 13 and 16 years found an internal reliability of Cronbach’s α = 0.82 and a high test–retest reliability of *r* = 0.91 [45].

SEL: This was measured using the Positive Youth Development Inventory [46], a 55-item questionnaire measuring five subscales of youth development: character, competence, confidence, connection, and contribution. The Positive Youth Development Inventory used a 4-point Likert scale for each question, ranging from (strongly disagree (1), disagree (2), agree (3) and strongly agree (4). Once the total score is determined, each score per question is evaluated by dividing the total score by the number of questions answered. Scores have ranged from a minimum of one to a maximum of four points per question. Additionally, sub-scale scores per question can be determined for each of the five previously mentioned sub-scales of youth development. The authors have reported a reliability coefficient of Cronbach’s α = 0.92 [46]. Other studies conducted in adolescents have reported acceptable psychometrics of this survey. An Indonesian version reported a reliability coefficient of Cronbach’s α = 0.97 [47], and an Iranian version found a Cronbach’s α and test–retest reliability ranging between α = 0.74 and 0.83 [48].

STEM Learning: The Attitude towards STEM Questionnaire, 6-12th grade edition [49], was used to determine the levels of interest within various components of STEM subject areas, education, and careers. This survey comprises eight questions on mathematics, nine questions on science, nine questions on engineering and technology, and additional sections on “21st Century Learning” and future interest in STEM courses. Based on the curriculum of the THINK program, we focused on questions pertaining to mathematics, science, and engineering and technology. Each question was scored using a 5-point Likert scale ranging from strongly disagree (1), disagree (2), neither agree nor disagree (3), agree (4), and strongly agree (5). A psychometric analysis of 2500 middle school students found an internal consistency reliability value of Cronbach’s α = 0.89 for the total survey, with values of 0.75 for the science portion, 0.89 for mathematics, and 0.77 for the engineering and technology portions. The test–retest reliability value was Cronbach’s α = 0.80 for the total survey, 0.70 for science, 0.89 for mathematics, and 0.77 for engineering and technology [50].

### 2.5. Data Analysis

All collected data were analyzed using the SPSS statistical package (version 27, IBM SPSS Inc., Armonk, NY, USA). Mean values and standard errors of the means (SEMs) were determined for all physical and physical fitness variables, as well as completed questionnaires and surveys. Normality for physical and physical fitness variables was determined via visual and numerical assessments, with the criteria for skewness and kurtosis being ±2.00. A paired samples *t*-test was conducted at baseline and at the program’s completion to measure changes in variables at the end of the THINK summer program.

## 3. Results

### 3.1. Subject Characteristics

Presented in Table 1 are the baseline subject characteristics. Approximately 56.1% of the sample were Latino/x, followed by 26.2 non-Hispanic White, and Black and Asian adolescents at 8.9% and 8.8%, respectively. Male participants comprised 55.3% of the sample, while females comprised 44.7% of the entire group. The distribution of grades for this sample was approximately 22% in 6th grade, 48% in 7th grade, and 30% in 8th grade.

### 3.2. Physical Characteristics

Table 2 illustrates the physical and anthropometric characteristics at baseline and completion of the 6-week THINK program. BMI, the percentage of participants at the 85th percentile for BMI, and total body fat percent did not statistically change from baseline to post-testing. The only anthropometric variable that changed was lean body mass, which significantly increased by 1.36 ± 0.72 kgs (*p* < 0.001).

### 3.3. Physical Fitness Measures

Table 3 summarizes the results of the physical fitness testing. Using the Pacer test, cardiorespiratory fitness increased, with attendees completing 3.39 (*p* < 0.001) more laps. Power improved by 2.7 cm (*p* < 0.05) using the Vertec, while flexibility increasing by 1.62 cm (*p* < 0.001) using the Sit-and-Reach test. By reducing the time to complete the shuttle run, agility improved by 0.31 s (*p* < 0.001) and muscular endurance improved, with an increase of 7.23 curl-ups (*p* < 0.001) completed in one minute.

### 3.4. Self-Reported Surveys and Questionnaires

Table 4 summarizes the self-reported surveys and questionnaires. Using the AFHC, food habits increased by 1.6 points (*p* < 0.001) and SEL using the Positive Youth Inventory improved by 0.05 points (*p* < 0.05). No significant changes were found in STEM learning.

## 4. Discussion

Despite the well-known advantages of regular physical activity, only 24.8% of youth meet national guidelines for physical activity participation [51]. In a previous study, 33.6% of adolescents were reported to score low on their cardiorespiratory testing—with non-Hispanic Black and Hispanic/Latino/x youth scoring lower than their non-Hispanic white peers [52]. Given that 65% of participants in our program were Hispanic and Black, the 13.6% improvement in cardiorespiratory fitness found in our study was especially meaningful. This could be related to the high levels of aerobic activity, the numerous team sports such as soccer, basketball, and flag football, that kept participants active and on the field throughout the activity. Students participated in sports demanding aerobic activity an average 3–4 h per day. Research has shown that even small improvements in cardiorespiratory fitness translate into large reductions in cardiovascular risk later on in adulthood [11,52,53]. Significant improvements in power, abdominal endurance, and agility translate into positive gains in neuromuscular function that are shown to track well into adulthood. Furthermore, neuromuscular benefits are associated with improvements in skeletal mass, bone density, and cognitive processing [11]. These elements may be key to improvements in functional independence and activities of daily living later in life [54]. Positive gains in these measures were likely due to creative relays, unique obstacle courses, and circuit training frequently implemented in this study. These were engaged in 1–2 h/day. Increases in range of motion are associated with reduced arterial stiffness, greater vascular compliance, and improved endothelial function [55]. Therefore, improvements in lower back and hamstring flexibility may have positive implications for cardiovascular health in the future. Both team and individual sports started with a group warm-up and ended with group stretching. Students also participated in yoga classes one time/ per week wherein they learned how to stretch correctly doing various yoga poses and also received information on the importance of stretching and proper relaxation techniques. Overall, participants improved in six out of seven physical fitness outcomes with no gains in body fat. These results are particularly noteworthy since low fitness levels in youth are considered a powerful predictor of cardiometabolic health [56,57] and premature cardiovascular disease [58].

A rising body of literature supports the fact that children and adolescents are at greater health risk during the summer months when reductions in physical fitness [59] and increases in body fat and body weight [60,61] are often reported. Termed the “structured day hypothesis”, activities are reported to be more well-organized and planned during the school year; therefore, youth are more protected at this time in contrast to the summer months when activities are less regimented and more autonomous [62]. Although research has shown that in youth 8–14 years old, gains in adiposity are most evident during pubertal development [63], there were no changes in adiposity following the program. Thus, no gains in adiposity is a positive finding since the average age of our participants was 12.05 years, indicating attendees were in the middle of puberty. Furthermore, most of our participants were Hispanic/Latino/x and non-Hispanic Black, wherein rates of obesity are highest [64]. In contrast to adiposity, lean body mass has been associated with greater health benefits. In a meta-analysis of youth, increases in lean body mass marked by muscular fitness were associated with increases in bone health and decreases in cardiovascular risk factors, insulin resistance, and inflammatory biomarkers [65]. Participation in various team sports, obstacle courses, Tug of War, and circuit training contributed to gains in lean body mass and physical fitness. These changes underscore the importance of active summer programs to improve physical fitness when health status in youth is reported to decline [62]. This is especially relevant for Hispanic/Latino/x youth who, at 56.1% of our attendees, are reported to possess some of the lowest rates of physical activity [66].

One of the unique aspects of this program was the interdisciplinary nature of this curriculum. Sport and fitness activities were woven into lecture information presented in nutrition and exercise science emphasizing STEM learning. The SEL curriculum included several modules centered around adopting a positive growth mindset, establishing and achieving healthy lifestyle goals, regulating emotions to attain favorable results, understanding personality tendencies to overcome challenges, and achieving a positive interface within the community setting. These modules were effectively delivered in both large and small group formats enabling campers to effectively engage in activities with their peers and counselors who promoted applied and shared learning. The SEL principles and strategies presented were then reinforced by counselors throughout the entire summer program.

SEL was also used to enable students to think more about healthy eating behaviors. Thus, when it came to discussing improvements in lunch and dinner meals, participants were asked to reflect upon barriers to healthy eating, communicating with family members about ways to make more informed and healthy decisions, and discussing healthy options that would improve the quality of their lunches during the program. In a study of 9280 children using the Healthy Eating Index [67], it was shown that eating habits were well below average in the United States. The researchers found that middle-school-aged youth (9–13 years) had lower scores for total fruit, dairy, and whole grain consumption (*p* < 0.05) than younger age groups (4–8 years) but higher scores than high school groups (aged 14–18 years). Therefore, improvements in nutritional habits during the middle school ages may be protective against declines in healthy dietary habits as youth get older. Since adolescence marks a key transition period from primarily parent-controlled eating to self-directed and/or peer-influenced eating, youth are just beginning to have more control and autonomy in their food selections [68]. Therefore, a 12.47% improvement in food habits during this critical transition period is very encouraging. Furthermore, minority youth and those of lower socioeconomic status may be more susceptible to poor nutrition habits than non-Hispanic white attendees and/or those of higher socioeconomic status [69]. Therefore, improvements in nutritional habits in just six weeks may reflect a positive trend toward developing healthy eating behaviors in those who stand to benefit most.

### 4.1. Limitations

It is necessary to acknowledge several limitations within this study. Although the recruitment of subjects was solicited from all middle school administrators across several South Florida counties, enrollment was based upon a first come, first serve basis. This may have diminished the number of underserved working parents who could not respond quickly or who were unable to drive their children to camp. Neither data on parental socioeconomic status nor the percentage of free and reduced lunches were available. Although this was a short-duration program, adolescents were in the middle of puberty. A control group of similar adolescents attending a more traditional summer program or no summer program should be added to determine whether significant changes were simply due to biological growth and physical maturation rather than the program itself. Increases in STEM learning were not observed. It may be that concepts in STEM education require more time to understand and absorb. In the future, a longer camp duration and/or more time devoted to STEM activities would be necessary to determine whether participants require more time to interpret and feel more comfortable learning these concepts. Finally, the data were not analyzed by gender. Due to sample size reductions in several dependent variables, there was not sufficient statistical power to adequately examine differences by gender. However, as the data base increases, it would be important to examine gender differences in a separate paper.

### 4.2. Practical Recommendations and Future Research Directions

Given the findings of this study, we would encourage other health promotion programs to consider utilizing an integrated approach that interfaces physical, psychological, and educational well-being into a novel program that promotes personal health. Programs that focus on only one of these domains may miss the synergistic benefits of an integrative holistic summer program. Future research should further evaluate the effectiveness of multi-dimensional health and fitness programs for longer periods of time using larger adolescent populations. Furthermore, an analysis of gender differences should be included in an independent study as the data base increases. Additionally, incorporating some of these educational practices during other times of the year may provide further health benefits and even mitigate decreases in physical fitness and increases in adiposity occurring during the more structured school year and less structured summer months. Including accelerometry data may offer new insight into how exposure to healthy initiatives may impact general trends in physical activity outside of the program. Expanding upon dietary elements may provide further context into the importance of positively influencing nutritional behaviors—such as providing healthy meals or menus, utilizing home-based dietary recalls, and including parental feedback during the discussion of different nutrition themes. Lastly, qualitative feedback and interviews from participants and their families may enrich the impact of more holistic, integrative programs on adolescents and their families in our communities.

## 5. Conclusions

An innovative 6-week program integrating physical, psychological, and educational well-being can improve physical fitness, body composition, SEL, and nutrition habits. Interdisciplinary interventions such as the THINK summer program lasting as short as six weeks may be an effective method of improving healthy lifestyle behaviors among adolescents during the summer months when such behaviors are expected to decline. Future research that includes a longer-duration summer program with a control group is warranted to determine whether results of the THINK summer curriculum are specific to the program content and materials—given the promising implications of these findings.

## Figures and Tables

**Table 1 nutrients-16-01838-t001:** Subject characteristics at baseline (*n* = 121).

Subject Characteristics	Baseline
Age (years)	12.05
Height (cm)	155.38 ± 0.87
Weight (kg)	54.47 ± 1.14
Race/Ethnicity (%)	
Black	8.9
Hispanic	56.1
Asian	8.8
White	26.2
Gender (%)	
Male	55.3
Female	44.7

Note: Mean values for age, weight, and weight are listed. Percentages by race and gender (both out of 100%) are categorized for the entire sample.

**Table 2 nutrients-16-01838-t002:** Physical characteristics at baseline and post-testing.

Characteristics	Baseline	Post-Testing	*p*-Value
BMI (kg/m^2^)			
(*n* = 108)	21.54 ± 0.41	21.42 ± 0.40	0.307
BMI z-score			
(*n*= 108)	0.000 ± 0.091	0.000 ± 0.096	0.716
BMI Percentile (%)			
≥85th	36.6%	34.8%	0.285
Body Fat (%)			
(*n* = 109)	26.00 ± 0.90	25.43 ± 1.17	0.452
Lean Body Mass (kg)	38.41 ± 0.72	39.77 ± 0.72	<0.001

Note: Data are presented as means ± standard error of the mean. *n* = 121 unless otherwise indicated. BMI, body mass index.

**Table 3 nutrients-16-01838-t003:** Physical fitness measures at baseline and post-testing.

Physical Fitness	Baseline	Post-Testing	*p*-Value
Strength (kg) (*n* = 102)	19.96 ± 0.88	19.94 ± 0.82	0.950
Cardiorespiratory Fitness (laps) (*n* = 107)	24.94 ± 1.34	28.33 ± 1.43	<0.001
Power (cm) (*n* = 103)	39.29 ± 1.14	41.99 ± 1.27	0.006
Flexibility (cm) (*n* = 103)	26.65 ± 1.05	28.27 ± 1.07	<0.001
Agility (s) (*n* = 103)	10.21 ± 0.19	9.90 ± 0.19	<0.001
Curl-ups (repetitions) (*n* = 103)	36.48 ± 1.16	43.71 ± 1.67	<0.001

Note: Data are presented as means ± standard error of the mean. Strength was reported using grip dynamometry. Cardiorespiratory fitness was assessed using the Pacer Test. Power was reported as maximum jump height using the VERTEC. Flexibility was recorded using a Sit-and-Reach Box Test. Agility was assessed using the Shuttle-Run Test. Curl-ups were measured using the One-Minute Curl-Up Test.

**Table 4 nutrients-16-01838-t004:** Self-reported surveys/questionnaires.

Self-Reported Surveys/Questionnairies	Baseline	Post-Testing	*p*-Value
AFHC (*n* = 110)	12.83 ± 0.53	14.43 ± 0.54	<0.001
STEM Learning (*n* = 103)	3.54 ± 0.06	3.53 ± 0.58	0.934
SEL (*n* = 107)	3.27 ± 0.03	3.32 ± 0.03	0.038

Note: Data are presented as means ± standard error of the mean for each scored survey. The AFHC was scored using the Adolescents Food Habits Checklist. Scores ranged from 0 to 23 for each healthy response recorded per completed question (out of 23 items). Science, Technology, Engineering, and Mathematics (STEM) learning was assessed using the The Students’ Attitudes Towards STEM Survey (Version 6-12th Grade). The average point per question was recorded based upon a 1–5 point Likert, scale with 1 being the lowest level of STEM interest and learning and 5 being the highest. One’s social emotional learning (SEL) was assessed via the Positive Youth Development Inventory. The average points per question were recorded using a 1–4 point Likert scale, with 1 reflecting the lowest level of SEL competency and 4 being the highest.

## Data Availability

The data presented in this study are available from the corresponding author on request. The data are not publicly available due to ethical reasons.

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
