# Peer review of "A Novel Summer Camp Integrating Physical, Psychological, and Educational Health in Youth: The THINK Program"

_nutrients, 2024, doi:10.3390/nu16121838_

Round 1

Reviewer 1 Report

Comments and Suggestions for Authors

The paper is well-written. Some comments to improve it:

General comment:

The authors examined numerous parameters before and after the camp but did not compare the findings to a control group, which is a significant limitation of the research.

Introduction:

How does obesity among American adolescents compare to adolescents in other Western countries? And how about in Florida compared to the rest of the USA?

Methods:

Was it a free camp? Did those same 108 children attend every three years, or was it a different group each year? What did they eat there? Was there a predetermined healthy menu? It seems there was more focus on physical activity than on nutrition.

Please add the questionnaires in the appendix.

Add the subheading "data analysis."

Results:

Please add subheadings.

Table 1 - What is the distribution of classes/ages?

Discussion:

Practical recommendations and directions for future research should be added.

Author Response

Introduction: 

We included information on obesity prevalence rates from different countries particularly in Western Europe. We also included obesity rates in Florida compared to the rest of the United States. We checked to make sure that wording, tense, and grammar were appropriate for this section. 

Methods: 

We commented on the distribution of participants and whether or not there were returning participants from previous years in the text. We also informed reviewers that students brought their own lunches daily to the camp. Although the THINK program was primarily a sport and fitness program, nutrition was strongly integrated throughout the program. For example, when discussing health-related lectures (i.e. the heart), information on the Mediterranean dietary plan and other nutritional considerations that would reduce heart disease risk were discussed. In fact, each health-related theme had a nutrition component embedded into the lecture. As you will see from the revised manuscript, we also included physical activities that reflected nutritional information. During the social emotional learning segments, barriers to healthy eating and small changes to lunch and dinner meals to improve health were discussed. Therefore, a great deal of time and emphasis was given to nutrition as we feel that both nutrition and physical fitness are inextricably related. 

The questionnaires were added in the appendices. The Data Analysis subheading was added to the methods section (2.5).  

Results

Subheadings were added to the results sections. These include: 3.1 Subject Characteristics, 3.2 Physical Characteristics, 3.3 Physical Fitness Measures, and 3.4 Self-Reported Surveys & Questionnaires. 

In the results section, 3.1 Subject Characteristics, we added the distribution of participants by grade into the text. 

Discussion

We added a very important and relevant paragraph on "Practical Recommendations and Future Research Directions".

We agree wholeheartedly with the review about having a control group. We tried very hard to get one, and in-fact found another summer camp that would allow us to evaluate their camp as a control group.  Our THINK program was a funded study, and unfortunately the granting agency would not hire additional assistants to help us evaluate the control group. Since both programs were offered at the same time, this meant we would have to pull all of our own assistants out of our study to evaluate the control group. Although we really wanted to do the control group, we simply had no assistants available to perform the baseline and follow up evaluations on a control group. We did ask our funding organization each year for additional assistants to evaluate a control group, and each year they refused. In the future, we are hoping to fund additional counselors to evaluate a control group since we feel this is so important. It is unfortunate that most summer programs in the literature do not have a control group because of funding limitations.

We really value the recommendations by this reviewer, as we felt it greatly enhanced the quality of the manuscript. 

Reviewer 2 Report

Comments and Suggestions for Authors

This manuscript describes a myriad of outcomes, from physical to educational, over three years' worth of data from a 6-week summer youth camp focused on improving social abilities and increasing physical activity and healthy eating.  Overall, the manuscript is clearly outlines the findings and does an excellent job of covering the statistics used in the analysis. However, the manuscript needs to be carefully reviewed for typos and verb tense inconsistencies.  In particular, the last paragraph on pg 5 repeats itself. 

Minor comments:

1.  Under 2.1, SEL is spelled out even though it is spelled out earlier in the manuscript.

2.  With the large number of outcomes, some type of adjustment multiple comparisons should be considered. 

3.  Table 5 state that it presents t-values for the regression models.  Betas from the models would be more interpretable for the reader.

Author Response

Minor Comments

  1. We went through the entire manuscript very carefully to evaluate typing mistakes, verb tense inconsistencies, and grammatical errors. We omitted the last paragraph on page 5, which repeated itself. Once we spelled out SEL in introduction, we subsequently abbreviated it in section 2.1 and throughout the rest of the manuscript. We checked that for other abbreviations as well.
  2. Because we conducted a pair-wise sample t-test versus an analysis of variance, our statistician indicated that adjustments for multiple comparisons would not be necessary. 
  3. We eliminated table 5 as requested by this reviewer and another reviewer as we felt this table was not relevant to the main objectives of the manuscript.

Reviewer 3 Report

Comments and Suggestions for Authors

The content of the paper is within the scope of the journal. The introduction provided adequate information and structure to set up the research question raised in the manuscript. However, the methods, results, and discussion sections have some limitations and aspects that can be improved. Therefore, some significant changes are needed before considering it for publication.

Some of the specific comments to possibly improve the manuscript are highlighted below:

1.     Some aspects of formatting should be corrected (spelling). Please correct what is pointed out in the body of the manuscript.

2.     All statistical symbols must be in italics (n, p).

3.     In the Materials and Methods section, measurement instruments, especially self-reported questionnaires/surveys, should be better described. Are they validated for the study population? What is the range of scores? What are the maximum and minimum values? I think the authors should provide more details on this.

4.     I don't understand why the authors used multivariate regression in statistical procedures. This study aimed to determine whether the THINK program could improve physical fitness, nutrition habits, SEL, and STEM education in a 6-week summer program covering three years. The objective was not to find predictors of body fat percentage or Lean body mass for the total sample or according to gender.

5.     The authors presented the results of the multivariate analysis in Table 5. Considering the study's objective, this table and its associated text should be omitted. Additionally, I had several concerns about the author's approach in this analysis, as they did not report statistical values related to the different multivariate models (Beta value, R2, etc.).

6.     In the note of Table 4, authors should limit themselves to describing the abbreviations of questionaries, levels of significance, and the maximum and minimum values of each dimension to better understand the results.

7.     The note in Table 1 should be corrected (see my comment in the notes I made in the manuscript).

8.     The discussion should be more focused on the results found.

9.     The most significant limitation of the study is the need for a control group. To explore your data further, didn't you consider exploring the results/changes according to gender?

Comments on the Quality of English Language

Some details (spelling, grammar, etc.) have been suggested for correction. Please see what is pointed out in the body of the manuscript.

Author Response

  1. We went through the entire manuscript at length and corrected spelling errors, typos, tense errors, and grammatical consistencies throughout the body of the manuscript.
  2. All statistical symbols were italicized. 
  3. We added additional information describing the self-reported questionnaires and surveys as well as measures of validity and reliability. We also included minimum and maximum values both in the text and in the tables. We believe this helps provide more information on each of our dependent measures. 
  4. We removed table 5 as suggested by this reviewer and another reviewer as we agreed that this analysis was not relevant to the main objectives of the manuscript. 
  5. Since we omitted the table, we did not address the presentation of the results of table 5. 
  6. In the notes of table 4, we limited the information to spelling out the abbreviations, reporting the levels of significance, and the minimal and maximal values of each questionnaire/survey to better explain the results.
  7. We corrected the note in table 1 and put % next to race/ethnicity and gender information to appropriately address the units used for each. 
  8. We spend a good amount of time reviewing the discussion. We provided information on the our results and what the program entailed that may have reflected these results. We added a description of physical activities that were integrated into our nutritional lectures and laboratory based activities that were added to the discussion of our health-related themes. Since we found so many significant results on physical fitness variables, we felt it important to describe several of the activities included that may have contributed to the positive findings in physical fitness variables and nutrition habits. We also incorporated more information on how SEL was integrated into the nutritional information presented. Again, we felt that the infusion of behavioral (SEL) changes to our nutrition information in the lectures may have promoted positive changes in nutritional habits. 
  9. We wholeheartedly agree with the comment by this reviewer as well as the other reviewer about having a control group. We tried very hard to get a control group, and in-fact found another summer camp that would allow us to evaluate their camp as controls. Our THINK program was a funded study, and unfortunately the granting agency would not hire additional assistants to help us evaluate a control group. Since both programs were offered at the same time, this meant we would have to pull all of our assistants out of our own study in order to evaluate the control group. Although we really wanted to do the control group, we simply had no assistants available to perform the baseline and follow up evaluations on a control group. We did ask our funding organization each year for additional assistants to evaluate a control group, and each year they refused. In the future, we are hoping to fund additional counselors to evaluate a control group since we feel this is so important. It is unfortunate that most summer programs in the literature do not have a control group in their research because of funding limitations.

We wish to thank this reviewer and all 3 reviewers for their time, insightful comments, and beneficial suggestions as we felt their feedback really improved the quality and comprehensiveness of this manuscript. 

Round 2

Reviewer 1 Report

Comments and Suggestions for Authors

The authors addressed all the comments and improved the paper.

Now it is suitable for publication.